# Exploring Brain Dynamics Within the Approach–Avoidance Bias

**DOI:** 10.3390/brainsci15121276

**Published:** 2025-11-27

**Authors:** Aitana Grasso-Cladera, Johannes Solzbacher, Debora Nolte, Peter König

**Affiliations:** 1Institute of Cognitive Science, Osnabrück University, 49090 Osnabrück, Germany; jsolzbacher@uni-osnabrueck.de (J.S.); debora.nolte@uni-osnabrueck.de (D.N.); 2Department of Neurophysiology and Pathophysiology, Center of Experimental Medicine, University Medical Center Hamburg-Eppendorf, 20251 Hamburg, Germany

**Keywords:** approach–avoidance bias, neural dynamics, EEG, approach–avoidance task, event-related potentials, frontal alpha asymmetry

## Abstract

Background: Approach–avoidance behaviors are fundamental mechanisms guiding our interactions with the environment, driven by the emotional valence of stimuli. While previous research has extensively explored behavioral aspects of the AAB, the neural dynamics underlying these processes remain insufficiently understood. Objectives: The present study employs electroencephalography (EEG) to systematically investigate the neural correlates of AAB in a non-clinical population, focusing on stimulus- and response-locked event-related potentials (ERPs). Methods: Forty-three participants performed a classic Approach–Avoidance Task (AAT) while EEG activity was recorded. Results: Behavioral results confirmed the AAB effect, with faster reaction times in congruent compared to incongruent trials, as well as for positive versus negative trials. ERP analyses revealed significant differences in the *Valence* factor, with early effects for stimulus-locked trials and late differences at the parietal-occipital region for response-locked trials. However, no significant effects were found for the *Condition* factor, suggesting that the neural mechanisms differentiating congruent and incongruent responses might not be optimally captured through EEG. Additionally, frontal alpha asymmetry (FAA) analyses showed no significant differences between conditions, aligning with the literature. Conclusions: These findings provide novel insights into the temporal and spatial characteristics of AAB-related neural activity, emphasizing the role of early visual processing and motor preparation in affect-driven decision-making. Future research should incorporate methodological approaches for assessing AAB in ecologically valid settings.

## 1. Introduction

Approach–avoidance behaviors have been conceptualized as automatic responses generated by the organism in response to subjective assessments of environmental components (e.g., objects, events, agents) [1,2]. Generally, the evaluation is described in terms of the emotional valence of the cue [3,4,5,6], where positive or beneficial cues are more likely to be approached, and adverse or negative ones are avoided [7,8,9,10]. Empirical evidence supports the automaticity of these responses, demonstrating faster and more accurate reactions to stimuli congruent with the previously mentioned organism’s predispositions, which is known as the Approach–Avoidance Bias (AAB) [4,11,12,13,14,15]. Moreover, the embodied nature of AAB underscores its evolutionary significance, with physiological mechanisms guiding these behaviors in realistic, dynamic contexts [8,16]. This interplay between emotional valence, automaticity, and embodiment highlights the AABs’ evolutionary and adaptive role, offering valuable insights into how organisms navigate and respond to different environmental components.

Research into the AAB has largely prioritized behavioral dynamics, often focusing on reaction time differences as a key metric, while only a few studies have explored the neural mechanisms underlying these behaviors. Studies with humans and non-human animals have identified the structural and functional networks underlying approach and avoidance behaviors [17,18,19,20]. Within these networks, the prefrontal cortex (PFC), particularly the anterior (aPFC) and adjacent ventrolateral areas, shows greater activity during incongruent movements compared to congruent movements [21,22,23,24]. This higher activity is linked to structural connections with the amygdala and its role in emotional processing and regulation [25,26], as well as the coordination of diverse cognitive processes (e.g., task-switching, control processing; [27]). For the AAB, the PFC overrides automatic responses for incongruent reactions [21]. The evidence highlights the pivotal role of the PFC in the cognitive processes underlying the AAB through its interplay with the amygdala.

Moreover, approach–avoidance behaviors can be understood as the result of the dynamic interplay between explicit and implicit information processing [28,29,30]. Explicit processing, such as under incongruent conditions in the Approach–Avoidance Task (AAT), involves deliberation and reflective reasoning. This type of processing is effortful, slower, and relies on cognitive control functions mediated by cortical frontal brain regions [28]. In contrast, implicit processing, evident in congruent AAT conditions, is driven by emotion-based information processing, influencing response timing and involving subcortical structures associated with emotional processing, such as the amygdala and ventral striatum [28,31,32]. According to Parsons et al. [33] and Loijen et al. [28], flexible and adaptive socio-emotional behaviors arise from the ongoing interaction between implicit and explicit processing mechanisms, including those underlying approach–avoidance behaviors. This framework raises questions about the temporal dynamics of these processes and their investigation through diverse methodological approaches, such as electroencephalography (EEG). Ultimately, approach–avoidance behaviors reflect a balance between implicit, emotion-driven, and explicit, reflective mechanisms, where this interplay determines both the timing and nature of responses.

Although significant progress has been made in exploring the neural pathways associated with the AAB, inconsistent results in both temporal and spatial dynamics [34,35,36,37] have hindered a comprehensive understanding of the neural dynamics underlying evaluation, motivation, and decision-making processes. Furthermore, many articles exploring brain activity have focused on clinical populations [35,36,38,39]; however, the neural dynamics within the AAB in healthy subjects are not yet fully understood. The present exploratory study uses EEG to systematically examine brain dynamics in healthy participants performing a classic Approach–Avoidance Task (AAT), focusing on Event-Related Potentials (ERPs) and changes in frontal alpha synchronization (Frontal Alpha Asymmetry; FAA). To this end, we partially replicated the experimental paradigm implemented by Solzbacher and colleagues [16] and collected EEG data while participants performed the AAT. Following this approach, we aim to study cortical processes related to emotional decision-making by using EEG data. To our knowledge, this is the first study to systematically explore neural differences associated with the AAB using both stimulus- and response-locked trials in a non-clinical population. By systematically investigating differences between stimulus- and response-locked ERPs, we aim to deepen our understanding of human behavior by providing insights into the timing and nature of cognitive and emotional processes underlying the AAB.

## 2. Methods

### 2.1. Participants

A total of 43 participants (26 males) aged 19 to 36 years (mean age = 25.30, SD = 3.57) completed the experiment. All participants were right-handed, had normal or corrected-to-normal vision, and were advanced or native speakers of English. Before the experiment, written informed consent was obtained from all participants. They received course credit as compensation for their participation. All instructions were provided and explained in English. The study was approved by the Ethics Committee of Osnabrück University.

### 2.2. Stimuli

The stimulus set included 87 full-color pictures belonging to the International Affective Picture System [40], chosen to ensure comparability and consistency with prior research [15,16,41]. The images were selected considering the cumulative evidence and their reliability for evoking emotional states across diverse populations and contexts [42,43,44,45] and based on ratings of their Valence (pleasant vs. unpleasant) from the Self-Assessment Manikin (SAM) scale. Of the 87 pictures, 44 had valence ratings below three points (unpleasant stimuli; pictures IDs: 2053, 2301, 2345.1, 2456, 2751, 3181, 3230, 3300, 6350, 6540, 6550, 6560, 6563, 6838, 9000, 9075, 9140, 9184, 9220, 9250, 9254, 9332, 9340, 9342, 9414, 9419, 9424, 9427, 9520, 9530, 9560, 9630, 9800, 9810, 9830, 9832, 9900, 9902, 9905, 9908, 9909, 9925, 9940, 9941); the other 43 had valence ratings above seven (pleasant stimuli; pictures IDs: 1340, 1410, 1500, 1540, 1600, 1610, 1620, 2040, 2057, 2058, 2075, 2150, 2158, 2160, 2170, 2208, 2260, 2274, 2299, 2300, 2306, 2314, 2332, 2340, 2341, 2345, 2347, 2360, 2387, 2388, 2391, 2392, 2398, 2540, 2598, 4599, 4614, 4628, 4640, 4641, 5200, 5202, 5210, 8497). The pictures featured diverse content and were displayed at their original resolution (1024 × 768 pixels), centered against a gray background (RGB values: 182/182/182).

### 2.3. Apparatus

The stimuli were presented on a 24-inch LCD monitor (BenQ XL2420T; BenQ, Taipei, Taiwan) positioned 80 cm in front of the participants. The display was set to a resolution of 1920 × 1080 pixels with a refresh rate of 114 Hz. The experiment was implemented using Psychtoolbox V3 [46] in Matlab R2016b (MathWorks Company, Natick, MA, USA). Behavioral data were collected through a Logitech Extreme 3D Pro joystick, selected for its capacity to facilitate body-related gestures. Participants used the joystick to simulate approach–avoidance behaviors by pushing and pulling, allowing the setup to examine embodied elements of the automatic approach–avoidance bias [16]. Consistent with previous studies, reaction times were measured from the onset of the joystick’s push or pull motion [16,47].

Brain activity was collected using EEG. A 64-channel Ag/AgCl electrode system with a Waveguard cap (ANT, Hengelo, The Netherlands) and a Refa8 amplifier (TMSi, Oldenzaal, The Netherlands), managed through the asa-lab acquisition server (v4.9.4) on a Dell laptop (Dell Inc., Round Rock, TX, USA; Windows 7, 32-bit; Intel(R) Core(TM) i5-3320M CPU). Data were sampled at 1024 Hz and recorded using common reference. A ground electrode was positioned under the left collarbone. EOG electrodes were positioned above and under the left eye. Impedances were kept below 10 kΩ. Triggers for stimulus presentation and movement onset were sent using a parallel port.

### 2.4. Procedure and Design

The experimental design partially replicates a previous study by Solzbacher and colleagues [16]. Participants were seated in front of a computer screen and presented with a series of pictures displayed one at a time. From the different variations in the approach–avoidance tasks described in the literature, we opted to implement explicit instructions (i.e., active evaluation of the valence of the stimulus) since they appear to yield the most consistent results [12,48]. Hence, participants were instructed to classify the pictures as either positive or negative. This classification was made by performing a joystick movement either toward their body (i.e., a pull movement) or away from their body (i.e., a push movement). We followed this operationalization since it is widely employed in the literature and allowed us to reproduce a previous experiment. Nonetheless, we acknowledge the ongoing debate about the symbolic meaning of these movements, as addressed in detail by Grasso-Cladera et al. [49]. All participants performed the task with their dominant hand and were asked to respond to each picture as quickly and precisely as possible. It was not possible to rectify and correct response mistakes, and no feedback was delivered.

The task consisted of two blocks with different instructions (congruent or incongruent). The order of the blocks was randomized for each participant. In the congruent block, participants were instructed to react to positive pictures by performing a pull movement towards their body (approach) and to react to negative pictures with a push movement away from their body (avoid). In the incongruent block, the participants were instructed to approach the negative images and avoid the positive pictures. To ensure participants remained aware of the current block’s instructions, reminders were displayed every 20 trials.

Each trial began with a fixation cross displayed at the center of the screen for 3 s (±0.5 s). Then, the picture appeared and remained on the screen until the participant moved using the joystick. At the end of each trial, there was a pause of 0.5 s. Participants were presented with 40 images of different emotional valence, equally distributed, in each block. At the beginning of each block, the first four images were test trials and were excluded from the analysis. Each image was presented once during the entire experiment. The order of the stimuli was randomized between blocks and subjects.

Following previous studies [15,16], a “zoom effect” was incorporated to augment the impression of an approach or avoidance behavior. The “zoom effect” consists of dynamically altering the image sizes in response to joystick movements—images increase in size during approach (pull) movements and decrease in size during avoidance (push) movements. This feature started when the participants initialized the movement of the joystick and continued enlarging or reducing the image’s size at a fixed speed. The “zoom effect” was programmed in MATLAB’s Psychtoolbox V3 (r2017a; MathWorks Company; adjusted from Czeszumski et al. [15]). Figure 1 provides a graphical representation of the task.

Before the picture trials, participants performed 20 no-stimulus trials in which they were instructed to push or pull the joystick whenever the fixation cross changed color. This was performed to obtain baseline data while using the joystick and to help participants familiarize themselves with the movement.

## 3. Preprocessing

### 3.1. Preprocessing of Behavioral Data

The collected data was divided into two conditions: congruent and incongruent. We visually inspected the reaction time data, indicating a positive skew (i.e., right-skewed distribution) for both conditions. Such a distribution shape is characteristic of reaction-time experiments, where a concentration of faster responses and a tail of slower responses are commonly observed [50,51]. To address outliers, we first implemented a filtering procedure. Since the minimum reaction time for processing visual stimuli is typically around 200 ms [52,53], we identified all trials with reaction times below 150 ms as unintentional responses, excluding 16 trials across participants (4 in the congruent condition and 12 in the incongruent condition). These rapid responses were likely unrelated to the task and stimulus processing, potentially arising from factors such as the influence of hand position on the joystick, where resting hand weight may have hindered a return to the neutral position. Furthermore, in some trials, participants exhibited reaction times exceeding two standard deviations from the mean, resulting in a longer “tail” distribution. We applied a 2-standard-deviation Winsorizing process to adjust the dataset’s distribution while preserving valuable information. The 2-standard-deviation adjustment affected 112 trials across participants, representing 3.38% of the entire dataset (55 trials in congruent and 57 in incongruent conditions). Finally, participants with performance lower than 90% accuracy (*n* = 2) were excluded from further behavioral and EEG analyses.

### 3.2. EEG Data

EEG data preprocessing was conducted in the MATLAB (R2024b) environment using EEGLAB [54] (version 2024.1) with a custom script adjusted from Schmidt and Nolte [55]. The implemented preprocessing routine was developed following advanced preprocessing guidelines for EEG data and previous work in the field [56,57,58]. First, the data was imported into MATLAB, ensuring double precision for all preprocessing steps [59]. Non-empirical segments (e.g., pre-task intervals) were removed. Channel labels were standardized to the 10-5 BESA system, and channels with no recorded data were excluded.

We applied a Hamming window [60] and filtered the data with a low-pass filter at 128 Hz and a high-pass filter at 0.5 Hz using the *pop_eegfiltnew* function. We limit the frequency at 0.5 Hz, so the time constant of the filter would be short in comparison to the inter-trial distance. Subsequently, the data was downsampled from 1024 Hz to 500 Hz, after which we applied the Zapline Plus plugin [61,62] to attenuate line noise, specifically addressing spectral peaks around 50 Hz. Data was then re-referenced to the average to establish a consistent baseline.

The *clean_rawdata* function was used to automate cleaning of noisy channels (mean = 3.674, std = 2.53, min = 0; max = 9) and segments (mean = 5.413, std = 6.57; min = 1, max = 32) [63]. Given the higher-than-expected noise level, we applied a conservative burst criterion of 20, representing the standard deviation threshold for burst removal via Artifact Subspace Reconstruction (ASR). Following noisy channel removal, the data was again re-referenced to the average.

Independent component analysis (ICA) was conducted using the Adaptive Mixture of Independent Component Analyzers (AMICA) plugin (version 15) [64] to isolate and remove components associated with muscle, ocular, cardiac, and residual line or channel noise. For ICA preprocessing, a temporary high-pass filter at 2 Hz was applied to improve component estimation [65] Independent components (ICs) with greater than 80% muscle-related activity or over 90% other noise, as identified by ICLabel [66], were automatically rejected (mean = 7.372, std = 4.14, min = 2, max = 18). Then, EEG sensor-level data was reconstructed from the ICs and robustly identified as brain-related. Missing channels were interpolated using spherical interpolation. This procedure was implemented consistently across all subjects, ensuring uniform preprocessing before statistical analysis. Participants with less than 90% valid trials (*n* = 1 for stimulus-onset trials and *n* = 2 for movement-onset trials) were removed from further analyses.

Finally, we implemented a linear model using the Unfold toolbox [67] to correct for the effect of overlapping events due to the experimental design. In the model, we use both picture onset and movement onset as event factors, with the levels of the 2 × 2 design (i.e., Congruent—Positive, Congruent—Negative, Incongruent—Positive, Incongruent—Negative). The overlap correction adjustment was applied for −800 to 1000 ms surrounding picture onset and −800 to 800 ms surrounding movement onset.

## 4. Analyses

### 4.1. Behavioral Analysis

We employed a linear mixed model (LMM) to analyze the behavioral data, which assumes that the data follow an underlying normal distribution. Since our reaction time data exhibits an ex-Gaussian distribution [68], we applied a base-10 logarithmic transformation (log_10_) to the data to meet the normality assumption. We performed all subsequent behavioral analyses on the log-transformed reaction time data. The LMM was implemented in MATLAB using the fitglme function, with Maximum Pseudo-Likelihood (MPL) as the fitting method. We assumed constant degrees of freedom for the model. Effect sizes were computed using Cohen’s d [69,70,71], calculated by dividing the coefficient of each prediction by the corresponding standard error. We model reaction times by the condition (i.e., congruent/incongruent) and the emotional valence of the picture (i.e., pleasurable/unpleasurable) factors, which were defined as fixed effects. We tested the interactions between the mentioned factors. All predictors were included using the effect coding scheme [72]. Furthermore, we included the grouping variables subject and picture as random effects to account for variability arising from individual participant differences and picture-specific factors not explained by the valence rating. Equation (1) presents the Wilkinson notation of the implemented model.***Reaction Time ~* 1 + *Condition* * *Valence + (*1|*Subject)* + *(*1*|Picture)***(1)

### 4.2. EEG Analysis: Time Series Domain

To explore differences across conditions at all electrodes and time points, we performed a two-factor repeated measures ANOVA (2 × 2: *Condition* × *Valence*), with a significance level set at 0.05. To address the multiple comparison problems, we implemented a cluster-based permutation test using threshold-free cluster enhancement (TFCE) as described on the *ept_TFCE* toolbox [73]. We adopted this methodological strategy given the exploratory nature of our study, prioritizing a robust and unbiased data-driven approach [73]. We performed 10,000 permutations, randomizing the data across all factors and levels for each permutation, and then applied a two-factor repeated measures ANOVA. Subsequently, the resulting F-values were enhanced using TFCE with parameters E = 0.666 and H = 1, following recommendations for F-statistics [73]. This process generated an empirical null distribution of TFCE-enhanced F-values, derived by recording the maximum F-value across all channels and time points for each permutation. Observed TFCE-enhanced F-values were then evaluated against this null distribution, with statistical significance identified as values exceeding the 95th percentile of the null distribution.

### 4.3. EEG Analysis: Frontal Alpha Asymmetry (FAA)

The EEG signal’s power spectral density (PSD) was computed using Welch’s method to estimate the signal power distribution across frequencies. The activity post-picture onset of each epoch was analyzed individually by applying a Hamming window of 1 s duration (500 samples) with 50% overlap to mitigate spectral leakage. The PSD for each epoch was calculated using a 1024-point Fast Fourier Transform (FFT), resulting in a frequency resolution of approximately 0.49 Hz. The PSDs were averaged across all trials to obtain an average spectrum. Alpha band power (8–12 Hz) was extracted by integrating the PSD over the specified frequency range. This approximation provides a robust estimate of the spectral content while accounting for inter-epoch variability. Then, we calculated the FAA by subtracting the frontal activity on the alpha band of the right and left hemispheres [34,74]. We computed the FAA over single electrodes and left-right clusters. Finally, we implemented non-parametric statistics (Wilcoxon Signed-Rank test) to evaluate differences between pleasant and unpleasant trials.

## 5. Results

### 5.1. Behavioral Data

First, we investigated whether the collected behavioral data reproduces previously observed patterns of reaction time data as a function of condition and valence. For this, we applied a Linear Mixed Model. Interestingly, the results from the model revealed a significant main effect of Condition, t(3192) = −5.206, *p* < 0.0001. Specifically, the log-transformed reaction time data for the congruent condition was estimated to be lower than for the incongruent condition, with a coefficient of −0.008 (95% CI: [−0.0111, 0.0050]). This corresponds to the congruent condition having reaction times of approximately 0.9817 compared to those in the incongruent condition, derived from the exponential of the log-transformed coefficient (shown in Figure 2). This indicates that participants responded significantly faster in the congruent condition than in the incongruent condition, with a difference of approximately 1.83%. The effect size for the Condition factor was d = −5.2 (SE = 0.0015), corresponding to a large effect size [69]. Additionally, we found a significant main effect of Valence t(3192) = −2.95, *p* = 0.0032, which indicated that the log-transformed reaction time data for pleasant pictures was estimated to be lower than for unpleasant pictures, with a coefficient of −0.0118 (95% CI: [−0.0196, −0.0039]). This corresponds to the condition of the pleasant picture having reaction times of approximately 0.9732 than those in the condition of the unpleasant picture, derived from the exponential of the log-transformed coefficient (shown in Figure 2). Hence, participants performed a movement significantly faster for pictures rated as pleasant when compared to the unpleasant ones (2.68%; Figure 2). For the Valence factor, the effect size was d = −2.95 (SE = 0.0040), corresponding to a large effect size [69]. The results from the tested interaction of the relevant factors (Condition * Valence) showed no significant differences (*p* = 0.0652). Finally, the Interclass Correlation Coefficient was 0.4103 for the grouping variable Participants and 0.0825 for Pictures. Overall, the behavioral data results align with the AAB reaction time pattern.

**Figure 2 brainsci-15-01276-f002:**
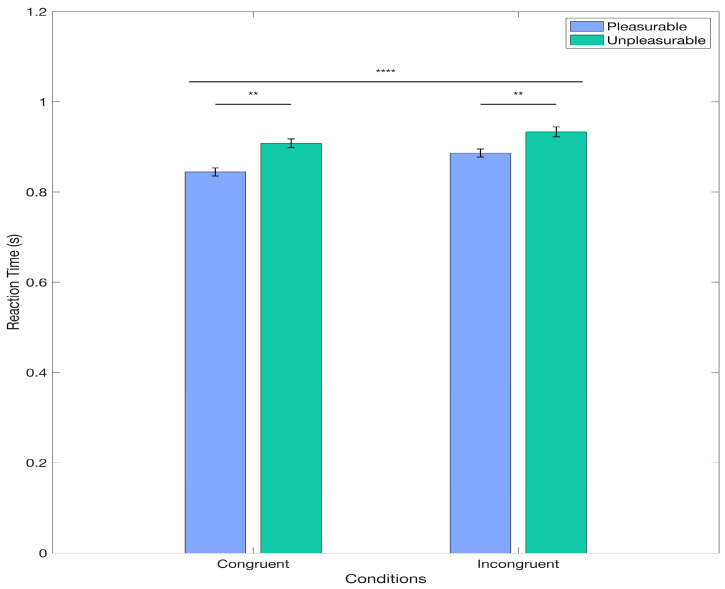
Main Effects for the LMM for Behavioral Data. The figure displays the mean reaction times for each condition and the standard error for the non-log-transformed data. Significant differences for the Condition (congruent vs. incongruent) and Valence (positive vs. negative) factors are displayed here. ** *p* < 0.05; **** *p* < 0.0001.

### 5.2. EEG Analysis: Time Series Domain

Regarding the time-series domain, we aimed to explore differences across conditions at all electrodes and time points for stimulus- and movement-onset ERPs. For this, we first preprocessed the data by implementing a linear model using the Unfold toolbox [67] to correct for the effect of overlapping events (stimulus and movement onset) due to the experimental design. Then we proceeded to epoch the data (−800 to 1000 ms for stimulus-onset ERPs and −800 to 800 ms for movement-onset ERPs). A total of 3060 trials were considered valid for stimulus-onset and 2987 for movement-onset. Considering the large number of electrodes and time points, we conducted a mass univariate analysis [75,76,77] to explore condition-specific differences along electrodes and time. We used threshold-free cluster enhancement (TFCE) [73] as a correction for multiple comparisons with a significance level at alpha < 0.05. Following this procedure, we were able to perform data-driven analyses of complex EEG data on the time-series domain.

We applied the described procedures for investigating different conditions (Congruent—Positive, Congruent—Negative, Incongruent—Positive, Incongruent—Negative) for stimulus- and movement-locked ERPs. We found significant differences in the Valence factor and the interaction of factors for both stimulus- and response-locked ERPs, and there was no significant cluster for the Condition factor. For stimulus-locked analyses, we found the peak for the significant cluster for the Valence factor at 86 ms after stimulus onset at channel M2, while for the interaction of factors, the peak was at 582 ms after stimulus onset at channel P6. The average ERP over conditions at M2 showed a highest peak of 3.056 μV at ~130 ms post picture onset, and a lowest peak (−0.778 μV) at ~180 ms post stimulus onset. At channel P6, the average over conditions showed a positive peak of 7.891 μV at ~125 ms after picture presentation, and a negative deflection (−4.398 μV) with a peak at ~20 ms post stimulus onset. For response-locked ERPs, we found the peak of the cluster for the Valence factor at 2 ms after movement onset at channel P5 and 108 ms post movement onset at channel POz for the interaction of factors. The average ERP over conditions at channel P5 presented a positive peak at ~−30 ms previous to movement onset (1.901 μV) and a negative peak of −2.034 at ~130 ms after movement onset. Similarly, at channel POz, the average ERP across conditions showed a positive deflection of 2.861 μV at ~−30 ms before movement onset, and a negative peak at ~130 ms after movement onset (−4.901 μV).

Figure 3 displays each significant cluster peak’s ERPs. The results from stimulus- and response-locked ERPs show significant clusters for the Valence factor and the interaction of factors, but not for the Condition factor.

The spatial distribution of the cluster’s peak varied for every ERP type and significant level. For stimulus-locked ERPs in the Valence factor, the peak of the cluster encompassed 36 electrodes distributed across frontal, central, and posterior areas (shown in Figure 4A), while for the interaction of factors, the peak of the cluster included 10 electrodes with a parietal-occipital distribution (shown in Figure 4B). Furthermore, for response-locked ERPs, the extent of the peak of the significant cluster was seven electrodes, with a parietal-occipital distribution (shown in Figure 4C), and for the interaction of factors, there were 50 electrodes included in the peak of the cluster distributed across different topographic areas (shown in Figure 4D). Overall, the topographical extension of the significant clusters varied for stimulus- and movement-locked ERPs and for the significant levels.

### 5.3. Stimulus-Locked Cluster Characteristics

In the following sections, we will describe the results, specifically the cluster extent, for the Valence factor and the interaction of factors for ERPs aligned to picture-onset, given that there are no differences for the Condition factor.

The mass univariate analysis showed differences in the Valence factor of the stimulus-onset ERPs. We performed a test for significant differences using TFCE (2 × 2 ANOVA, alpha < 0.05), which revealed a significant cluster highly compatible with an effect spanning from 30 to 126 ms post-picture onset (shown in Figure 5A–D), encompassing a total of 57 (AF3, AF4, AF7, AF8, C1, C2, C3, C4, C5, CP1, CP2, CP3, CP5, CP6, CPz, Cz, F1, F2, F3, F4, F5, F6, F8, FC1, FC2, FC3, FC4, FC6, FCz, FP1, FP2, FPz, Fz, M1, M2, O1, O2, Oz, P1, P2, P3, P4, P5, P6, P7, P8, PO3, PO4, PO5, PO6, PO7, PO8, POz, Pz, T8, TP7, TP8) out of the 64 electrodes. The peak of the cluster was found at channel M2, at 86 ms after picture onset (shown in Figure 5A and the second difference plot in 5E), presenting a median difference between positive and negative trials of −1.921 μV (std = 1.624, [−5.703 1.181]) of this time-electrode combination. The peak of this cluster is congruent with the P100 component, associated with early visual preprocessing. This result shows significant early differences after picture onset regarding the emotional valence of the stimuli; the difference is driven by higher amplitudes observed during negative trials than positive ones.

Similarly, the mass univariate analysis showed differences in the interaction between factors in the picture-onset ERPs. We found a significant cluster suitable with an effect spanning from 574 to 618 ms post-picture onset. The cluster encompassed a total of 15 (CP6, O1, O2, Oz, P4, P6, P8, PO3, PO4, PO5, PO6, PO7, PO8, POz, TP8) electrodes (shown in Figure 6). We found the cluster peak at channel P6, at 582 ms after picture onset (shown in Figure 6, third topoplot). Our results showed a late main effect due to differences between conditions in limited time and space windows.

### 5.4. Movement-Locked Cluster Characteristics

In the next sections, we will describe the cluster characteristics for the Valence factor and the interaction of factors for ERPs aligned to response-onset, given that there are no differences for the Condition factor.

Following the same procedure for the data epoched to stimulus-onset, we found differences in the Valence factor after performing a mass univariate analysis on the movement-onset epoched data. The analysis using TFCE (2 × 2 ANOVA, significance level < 0.05) showed a significant cluster for the Valence condition, corresponding with an effect spanning from −30 to 10 ms after movement onset (shown in Figure 7A–D), which included 12 (CP5, O1, O2, Oz, P1, P3, P5, P7, PO3, PO5, PO7, TP7) out of the 64 electrodes. The peak of the cluster was on electrode P5, at 2 ms after movement onset, F(38) = 30.1562, *p* = 0.0056 (shown in Figure 7C and the third difference plot in Figure 7E), presenting a median difference between positive and negative trials of −1.3 μV (std = 1.483, [−5.342 2.284]) of this time-electrode combination. Furthermore, topographical dynamics are maintained through the cluster temporal window. This result shows significant early differences after movement onset regarding the emotional valence of the stimuli; the difference is driven by higher amplitudes observed during negative trials than positive ones.

Furthermore, the mass univariate analysis showed differences in the interaction of factors on the movement-onset ERPs. We found a significant cluster suitable with an effect spanning from 28 to 312 ms post-movement onset encompassing all electrodes (shown in Figure 8). We identified the cluster peak at channel POz at 108 ms after picture onset, F(38) = 86.6177, *p* < 0.0001 (shown in Figure 8, third topoplot). The topographical dynamics are maintained through the cluster temporal window. Overall, our results showed a late main effect due to differences between conditions in a relatively limited time window but a large spatial distribution encompassing all electrodes.

**Figure 7 brainsci-15-01276-f007:**
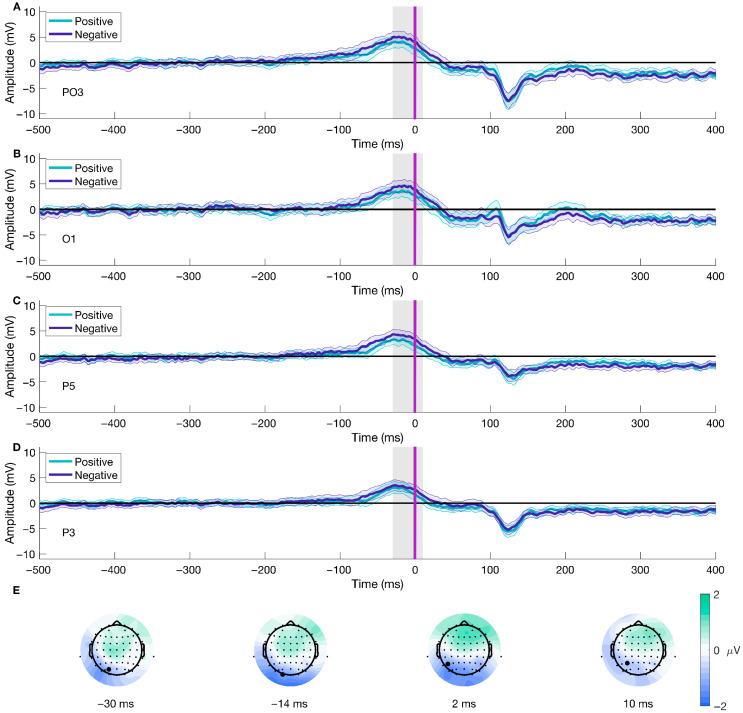
Movement-onset ERP for Valence and topographical distribution of the cluster over time. Panels (**A**–**D**) show the mean ERP locked to movement onset (purple line, Figure 1C) for different electrodes contributing to the cluster for both positive and negative trials. Panel (**E**) displays the average difference between conditions over time at four distinct time points. The gray line represents the temporal peak of the cluster. Highlighted electrodes on Panel (**E**) show the position of the ERP channels. Panel (**C**) and the third topographical map on Panel (**E**) are also shown in Figure 3 and Figure 4, respectively. We have shortened the temporal window to −500 to 400 ms for visualization purposes. The color bar represents the difference in voltage (in microvolts) between conditions, with green scale colors indicating relatively more positive differences and blue colors indicating more negative differences.

**Figure 8 brainsci-15-01276-f008:**
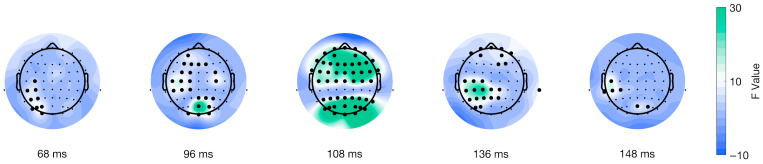
Topographical distribution of F values for the movement-locked interaction of factors as a function of time. The different topographical plots show the distribution of F values from the TFCE analysis over time for ERPs locked to movement onset (purple line in Figure 1C). Highlighted electrodes represent those that are part of the significant cluster at each time point. The color bar represents the F values, with green indicating higher values and blue indicating lower values.

### 5.5. EEG Analysis: Frontal Alpha Asymmetry (FAA)

We conducted non-parametric tests (Wilcoxon Signed-Rank test) to assess differences in FAA after stimulus-onset. First, we computed FAA on single electrodes (F4 and F3) and assessed the difference between pleasant and unpleasant trials, which resulted in no significant differences (*p* = 0.961). Then, we computed FAA averaging frontal electrodes surrounding F4 (AF4, F2, F6, F4, FC2, FC4, FC6) and F3 (AF3, F5, F1, F3, FC3, FC5, FC1) on the 10–20 positioning system, to obtain a cluster approximation for frontal-right and frontal-left electrodes as described in the literature [34,78]. As before, we assessed differences between pleasant and unpleasant trials, which showed no significant differences (*p* = 0.604). The results from the FAA analyses showed no significant differences in alpha asymmetry for pleasant and unpleasant trials.

## 6. Discussion

The present study aimed to systematically explore brain dynamics within the AAB. We adopted an exploratory approach because the existing literature presents conflicting findings, and due to the absence of a clear neural component (in the time domain) related to the AAB. Furthermore, this data-driven strategy also helps minimize confirmation bias, as the current state of knowledge does not provide sufficient grounds for a strong hypothesis-driven approach. To this end, we replicated the classic AAT setup implemented by Solzbacher and colleagues [16] and collected EEG data. Then, we analyzed the behavioral data and differences in the time-series domain and the FAA of the neurophysiological data. We found significant differences in behavioral and neurophysiological data, which we will demonstrate in the following sections to partially align with and extend upon previous findings in the field.

Our behavioral results show a main effect on the Valence and Condition factors, replicating findings described in the literature and supporting the existence of the bias as assessed by the AATs. We found that reaction times were shorter when participants interacted with a pleasurable rather than an unpleasurable valenced picture. Similarly, we found that reaction times were shorter when participants performed a movement congruent with the natural predispositions (e.g., pushing the unpleasurable pictures and pulling the pleasurable ones) than reaction times for the incongruent conditions. However, we did not find significant differences in the interaction of factors. These results align with the findings described by Solzbacher and colleagues [16] as well as in other studies [12,15,79] and support the assumptions about approach–avoidance behaviors. This reflects the evolutionary-rooted tendency to approach faster stimuli perceived as positive or beneficial rather than harmful or adverse ones, and to be faster when behaving congruently than incongruently regarding natural predisposition.

Furthermore, the time-series analyses of movement (purple line; Figure 1C) and picture onset (red line; Figure 1C) showed significant effects on the interaction of Condition and Valence and the Valence factor. Regarding the interactional effect, this result best describes the tendency related to the AAB (i.e., being faster to approach the positive and avoid the negative). It reflects the experimental design commonly implemented to study the bias. Hence, it proves the adequacy of the implemented behavioral task and analytical strategy for studying the neural dynamics of the AAB. For stimulus-locked trials (picture onset; Figure 1C), our results show a significant difference between conditions with a cluster from 574 to 618 ms (peak at 582 ms), with a parietal-posterior distribution. Given the average reaction times in our experiment, this cluster likely occurs before movement onset, falling within the period of decision-making and movement planning. Previous studies on the AAB have reported late differences in the interaction of factors that align in time and topography with the Late Positive Potential (LPP), reflecting enhanced motivated attention for emotional stimuli [36,37]. However, our results show a negative potential, which challenges common interpretations about the temporality and topographical distribution of the difference and the corresponding underlying cognitive processes. The negativity, as well as the spatial and temporal resolution of the difference, point out the direction of the readiness potential [80,81], highlighting the period before movement onset where the decision regarding the movement to perform based on the emotional valence and the given instructions is made [82,83,84]. Nonetheless, it is crucial to note that the trial alignment is to stimulus, not movement onset, and therefore this interpretation remains tentative. In contrast, response-locked trials (movement onset; Figure 1C), revealed a significant cluster after movement onset, ranging from 28 to 312 ms, with a maximum peak at 108 ms, located in the parietal-occipital region that might be functionally related to perceptual and evaluative processes. We would have expected earlier negative differences, mostly previous movement onset, as a marker of the readiness potential for preparation to action. The absence of such an early negativity, combined with the stimulus-locked findings, complicates a straightforward interpretation of readiness potential differences. Instead, we believe the difference we observe can be explained by the task’s properties, such as the stimuli’s zooming effect, which aligns with our results’ temporal and spatial characteristics. Future work can rule out this effect by comparing the data from two experimental tasks, with and without the addition of the zoom effect. To our knowledge, this study is one of the first attempts to systematically explore movement onset ERPs in the context of the AAB. Therefore, drawing definitive conclusions about the found clusters and their attributions to different processes, such as the readiness potential, is currently only speculative. Further research is needed to disentangle the contributions of valence to movement-related ERP differences. In summary, we found significant differences in ERPs aligned to picture and movement onset regarding the interactional factor. Our results for stimulus-locked trials most likely correspond to differences regarding movement preparation and decision-making, while the differences in movement-locked trials might point to task characteristics.

Investigating the influence of valence revealed significant differences in ERPs for negative compared to positive stimuli in trials aligned to picture onset (red line; Figure 1C), temporally and spatially corresponding to the P100 component. These findings are consistent with previous research reporting differences in the P100 amplitude associated with emotional processing [85,86,87], which is further supported by the posterior and parietal distribution of the cluster. These differences are believed to reflect the early stages of emotional processing [85,88], potentially priming the visual signal to timely responses [86,89]. Response-locked trials (purple line; Figure 1C) revealed a significant cluster centered around or shortly after movement onset, involving all electrodes, which suggests a global modulation and a joint cognitive response to the task. To our knowledge, our study is the first to investigate valence-related differences for movement-related ERPs in the AAB paradigm. The underlying neuronal processes contributing to the observed cluster are not identifiable based on our results alone. The cluster’s peak was contralateral, suggesting that early stages of movement execution likely contributed to the observed differences [90]. Moreover, the cluster extended to before movement onset, consistent with the readiness potential [80,81]. However, the positive rather than negative potential challenges its interpretation and the attribution to readiness potential differences. Further research is needed to disentangle the contributions of valence to movement-related ERP differences. Overall, we observed differences in response-locked and stimulus-locked trials for the valence factor. These results support the existence of early differences in the P100 component and are the first to investigate variations associated with movement execution.

Unexpectedly, our study did not find a significant effect of the Condition factor. Considering the relevance of deliberation and reflective reasoning processes based on cognitive control functions [28], we would have expected to find differences in brain activity when comparing congruent and incongruent conditions. This is mainly due to the high cognitive load of counter-intuitive decisions (e.g., approaching an unpleasant picture or avoiding a pleasant one) [91,92]. Furthermore, given the role of the PFC as part of both the underlying mechanisms for approach and avoidance behaviors, as well as for decision-making processes [17,18,19,20,21,22,24], we would have expected to see differences located in frontal regions either after picture onset or before movement onset, which will correspond to the deliberation or decision-making period. Even when some studies have reported differences in the Condition factor when analyzing ERPs in different contexts of the AAB [93,94,95], it might be that due to the characteristics of the primary structure involved, i.e., the amygdala [25,26,28,31,32], that a cortical analysis such as ERP or EEG, in general, is not a suited approximation and other techniques that can provide better insights for subcortical analyses should be considered (e.g., fMRI or functional Near-Infrared Spectroscopy; fNIRS). Similarly, two methodological aspects of the design might impact our results. First, the open temporal window for reaction time can include different cognitive processes, especially when considering the differences in reaction times (Behavioral Results Section). Secondly, the reduced number of trials (*n* = 20 per condition) might not be enough to elicit an ERP and account for the statistical power of the differences. While the current study did not find the expected effect of the Condition factor, employing different methodological approaches in future studies might better describe the brain dynamics related to the AAB.

In contrast to the time-series analysis, our results showed no significant differences between FAA values for pleasant and unpleasant trials for stimulus-locked trials. Higher left hemispheric activity tends to be associated with positive and pleasurable stimuli, while higher right activity has been associated with negative valenced stimuli [88,89]. Given the characteristics of FAA, it has been a preferred measure to explore activity related to affective stimuli and motivation [96,97]; nevertheless, results are heterogeneous. The review conducted by Sabu and colleagues [97] shows that out of 18 studies using emotionally valenced pictures to assess differences in FAA, only two found an effect of the presented stimuli [98,99]. However, studies implementing videos [100,101], real cues [102,103], and games [104,105] were more prone to finding FAA differences. These results highlight a relevant methodological point regarding the suitability of static images for generating emotional engagement in the subjects and eliciting differences in FAA activity, especially when considering the positive results found by implementing relatable real-world cues or more immersive experiences by using videos, games, and even 3D stimuli and virtual environments [49,97,106]. In this sense, the consistent finding of limited FAA effects with static images across multiple studies suggests a systematic influence of stimulus type. This consistent lack of effect when using static stimuli likely explains the absence of a difference in the current study, suggesting that FAA may not be the most suitable measure in classical AAB experiments using picture stimuli. Instead, the heterogeneity in FAA results regarding emotionally valenced stimuli, and the high prevalence of negative results when using static stimuli or 2D pictures, posit the need for new methodological approaches for studying the AAB under more emotionally engaging and naturalistic scenarios.

## 7. Limitations

While our study provides insights into the brain dynamics of the AAB, certain methodological and analytical constraints need to be considered. First, regarding the experimental design of the task, it is worth noting that our results are consistent with the standard view of the AAB as a distance-regulation mechanism, in which changes in proximity between an agent and an object underlie approach–avoidance behavior [11,16,49]. However, although pushing and pulling movements are commonly used to operationalize the bias in both desktop and VR tasks [9,16,107], their interpretation is not straightforward [108]. These actions do not inherently map onto approach or avoidance, as the same movement can serve opposite functions depending on context and stimulus properties [109,110,111,112]. To mitigate the effect of the symbolic meaning of pushing and pulling, we implemented a series of methodological considerations. First, the “zoom effect” (i.e., increasing the size of the picture when pulling and decreasing its size when pushing) after the movement performed by the subject served as a way to augment the impression of an approach or avoidance behavior. Furthermore, we repeated the presentation of the instructions after a regular number of trials, in order to prevent confusion regarding the type of block (i.e., congruent or incongruent) and the type of movement required for each valence in the current trial. Lastly, we incorporated a series of test trials at the beginning of each block as a way to generate familiarity with the task and the required movements for approaching and avoiding. Overall, regarding the open debate about the symbolic meaning of the movements used to study approach and avoidance behaviors, our experimental design attempted to mitigate its effect on the results.

Similarly, some limitations of the EEG data should be acknowledged to ensure a cautious interpretation of the results and to highlight directions for future work. To start, our experiment had a relatively small number of trials per condition (*n* = 20). Because EEG signals are inherently noisy, averaging across trials is essential to increase the signal-to-noise ratio (SNR) and obtain reliable estimates of ERPs [113,114]. Previous research has shown that the reliability of ERP measures increases substantially with trial count, particularly for components with lower amplitudes or more variable latencies [115,116]. With a limited number of trials, as in our case, residual noise may therefore contribute to increased variability across participants and reduced statistical power to detect effects [115,117]. Future studies should include more trials per condition to enhance SNR and improve the robustness and reproducibility of the results.

Furthermore, EEG, as a measurement technique, has lower spatial resolution in comparison with other neuroimaging techniques for the study of the AAB modulation on brain activity. This is especially relevant for the present study, since the primary structures involved in approach–avoidance tendencies are related to the limbic system and, therefore, are subcortical. Hence, the EEG measurement offers poor sensitivity to subcortical activity in regions associated with emotional processing and decision-making, as it can measure only cortical sources of activity. The interpretation of the present results must take these limitations into account, particularly when considering findings such as the absence of a significant effect of the Condition (e.g., congruent vs. incongruent). Although the data suggest that the different conditions do not modulate cortical activity detectable at the scalp level, this does not imply that no neural differences exist. Rather, such differences may simply not be accessible with EEG recordings. Given this limitation, future studies would benefit from incorporating complementary neuroimaging techniques (e.g., fNIRS and fMRI) to examine the brain dynamics of the AAB. Furthermore, incorporating other interventions along with EEG measurement, such as transcranial focused ultrasound (tFUS) [118,119,120], may provide deeper insights into the effects of the AAB on neural dynamics. Together, acknowledging the limitations of EEG and supporting these shortcomings with other strategies would advance the study and interpretation of the AAB modulation at the subcortical level.

## 8. Conclusions

Overall, this study systematically explored neural dynamics within the Attentional Bias (AAB) by implementing a classic AAT setup and collecting both behavioral and neurophysiological data. Our exploratory analysis revealed significant differences between the behavioral data and ERP findings, though no significant differences were found for the FAA. Our behavioral results align with existing literature on the AAB, which adds to the extensive body of evidence supporting the evolutionary mechanisms of the bias. In contrast, the ERP results offer partial support for previous studies, though the literature on the neural dynamics of the AAB remains varied and heterogeneous. Importantly, this study is the first to systematically assess differences across time and topographies for both stimulus- and response-locked ERPs, marking a significant contribution to the field. We aim for this work to serve as a foundation for further systematic investigations into the neural dynamics of the AAB, with the potential to deepen our understanding of these processes. The absence of significant differences in FAA results is consistent with a substantial body of prior research, suggesting that methodological factors may play a key role in shaping these outcomes. Overall, our findings underscore the complexity of the neural mechanisms underlying the AAB and highlight the need for more nuanced methodologies in future studies. Despite the variability in the ERP results, our study offers relevant insights into the temporal and topographical patterns of neural activity associated with the AAB. Further research should build upon these findings, refining experimental paradigms and employing more sophisticated analytical techniques to clarify the neural dynamics of attentional biases.

## Figures and Tables

**Figure 1 brainsci-15-01276-f001:**
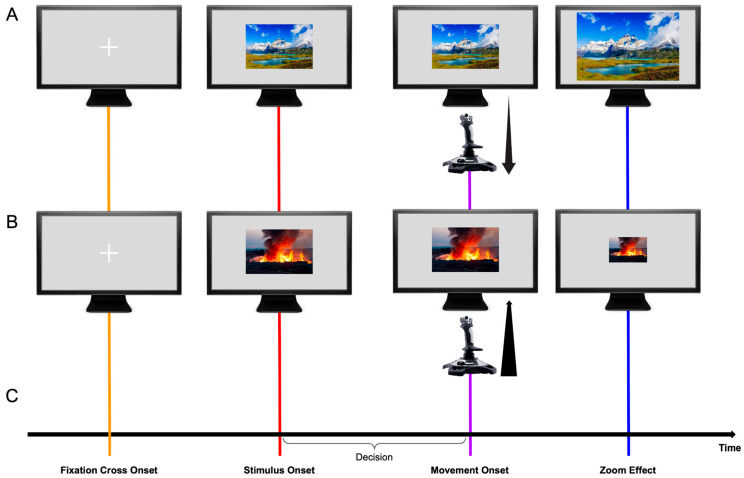
Graphical representation of the experimental paradigm (In order to maintain the experimental value of the IAPS collection, the images displayed in the figure are not part of it, but randomly selected pictures for the graphical representation of the experimental paradigm. Panel (**A**): Sriwongthai, (n.d.); Panel (**B**): Mulder, (n.d.).). Panels (**A**,**B**) show an example of a congruent block, where the participant performs a pulling movement with the joystick on a positive picture, which zooms in (**A**), or performs a pushing movement on a negative picture, which zooms out (**B**). The black arrows on Movement Onset represent a pulling (**A**) and a pushing (**B**) movement. Panel (**C**) depicts the temporal sequence of events within a trial. The trial begins with the presentation of a fixation cross (orange line), followed by the target picture (red line) after a random time interval. Following the decision period, participants execute a push or pull movement (purple line) as instructed, after which the zoom effect enlarges or reduces the picture size. The timing of the movement onset varies across trials and subjects, depending on the individual duration of the decision period. Lines red and purple will be implemented across the result section to denote ERPs locked to stimulus and movement onset, respectively.

**Figure 3 brainsci-15-01276-f003:**
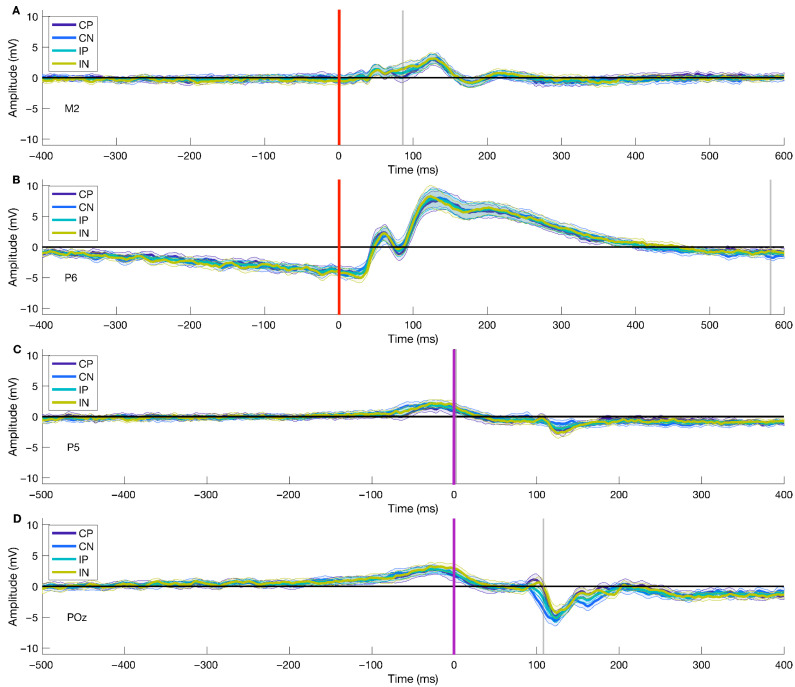
ERPs for all peaks of significant clusters. Panel (**A**) represents stimulus-onset ERPs (locked to picture onset; red line, Figure 1C) of all target conditions for the Valence factor, given its statistical significance. Panel (**B**) shows the stimulus-onset ERPs of all target conditions for the interaction of the Valence and Condition factors, given their statistical significance. Similarly, panels (**C**,**D**) display movement onset ERPs (locked to movement onset; purple line, Figure 1C) for Valence and the interaction of factors, respectively. The gray line represents the temporal peak of the cluster. For visualization purposes, we have shortened the temporal window to −400 to 600 ms for picture onset trials and −500 to 400 ms for movement onset trials.

**Figure 4 brainsci-15-01276-f004:**
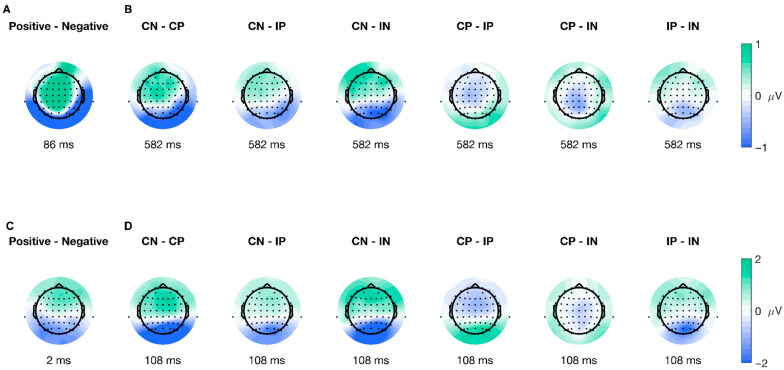
Topographic plots for all peaks of significant clusters. Panels (**A**,**B**) illustrate the average difference between conditions for the cluster peak on stimulus-onset ERPs (locked to picture onset; red line in Figure 1C), (**A**) for the Valence factor, and (**B**) for the interaction. Similarly, panels (**C**,**D**) show the average difference between conditions for the cluster peak on movement-onset ERPs (locked to movement onset; purple line in Figure 1C), with (**C**) displaying the Valence factor and (**D**) the interaction. The color bar represents the difference in voltage (in microvolts) between conditions, with green indicating relatively more positive differences and blue indicating relatively more negative differences.

**Figure 5 brainsci-15-01276-f005:**
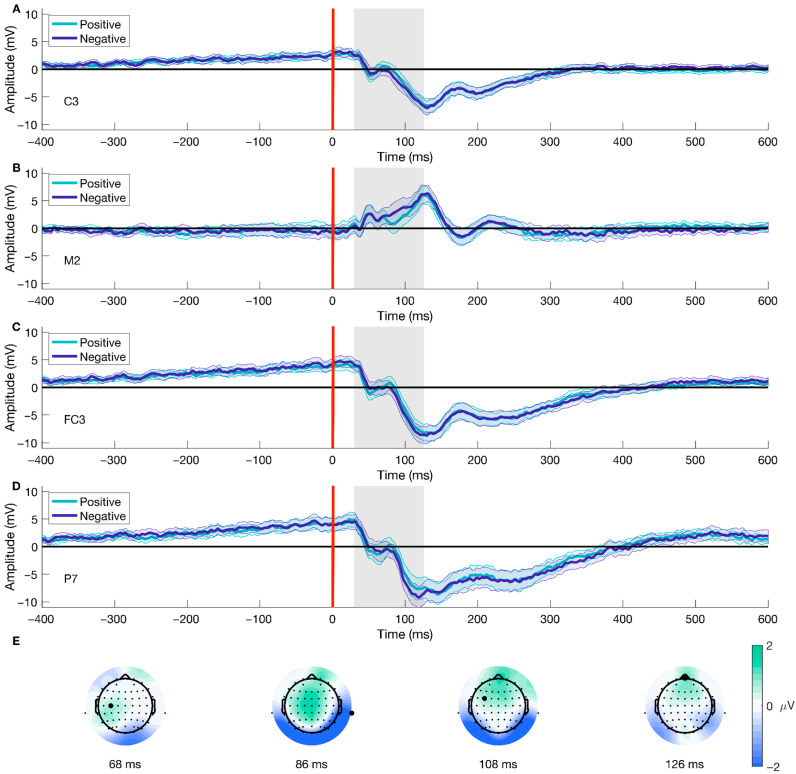
Stimulus-onset ERP for Valence and topographical distribution of the cluster over time. Panels (**A**–**D**) show the mean ERP locked to picture onset (red line, Figure 1C) for different electrodes contributing to the cluster for both positive and negative trials. Panel (**E**) displays the average difference between conditions over time at four distinct time points. The gray area represents the temporal extension of the cluster. The highlighted electrodes in Panel (**E**) indicate the positions of the ERP channels. Panel (**B**) and the second topographical map on Panel (**E**) are also shown in Figure 3 and Figure 4, respectively. We have shortened the temporal window to −400 to 600 ms for visualization purposes. The color bar represents the difference in voltage (in microvolts) between conditions, with green indicating relatively more positive differences and blue indicating more negative differences.

**Figure 6 brainsci-15-01276-f006:**
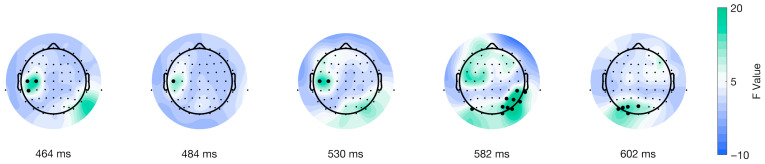
Topographical distribution of F values for the stimulus-locked interaction of factors as a function of time. The different topographical plots show the distribution of F values from the TFCE analysis over time for ERPs locked to picture onset (red line in Figure 1C). Highlighted electrodes represent those that are part of the significant cluster at each time point. The color bar represents the F values, with green indicating higher values and blue indicating lower values.

## Data Availability

The anonymized raw data is publicly available on the Open Science Platform: https://osf.io/4fmw9/ (accessed on 1 October 2025). The code used for the analysis of the present article is available at: https://github.com/AitanaGrasso-Cladera/Exploring-brain-dynamics-within-the-Approach-Avoidance-Bias (accessed on 1 October 2025) (Grasso-Cladera, A. & Nolte, D. Exploring-brain-dynamics-within-the-Approach-Avoidance-Bias (Version 1.0) [Computer software]. https://github.com/AitanaGrasso-Cladera/Exploring-brain-dynamics-within-the-Approach-Avoidance-Bias (accessed on 1 October 2025)).

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
