# Peer review of "Exploring Brain Dynamics Within the Approach–Avoidance Bias"

_brainsci, 2025, doi:10.3390/brainsci15121276_

Round 1
Reviewer 1 Report
Comments and Suggestions for Authors
The manuscript presents a well-designed study that systematically explores neural correlates of the Approach–Avoidance Bias (AAB) using both stimulus- and response-locked ERP analyses. The methodological rigor is commendable, and the results contribute novel insights into the temporal and spatial dynamics of AAB-related neural activity in non-clinical populations. However, several aspects could be clarified or refined to strengthen the theoretical and methodological interpretation of the findings.
Introduction
Lines 74–76: “Ultimately, approach–avoidance behaviors reflect a balance between explicit, emotion-driven, and implicit, reflective mechanisms…”
It seems that the terms explicit and implicit have been reversed here. According to earlier lines, explicit processes are reflective and deliberative, whereas implicit ones are emotion-driven and automatic. I suggest correcting this sentence for conceptual consistency.
Methods
Lines 139–140: “Nonetheless, we acknowledge the ongoing debate about the symbolic meaning of these movements, as addressed in detail by Grasso-Cladera et al. [45].”
This is an important conceptual limitation. It could be discussed in more depth in the Methods or Limitations section, clarifying how the authors handled or mitigated the symbolic ambiguity of push–pull movements (e.g., by including the zoom effect, explicit instructions, or task familiarization).
Lines 176–179: “Before the picture trials, participants performed 20 no-stimulus trials in which they were instructed to push or pull the joystick whenever the fixation cross changed color. This was done to obtain baseline data while using the joystick and to help participants familiarize themselves with the movement”
The paragraph appears twice. Please remove the duplicate section.
Discussion
Lines 475–479:
The authors propose that the observed negative potential may reflect a readiness potential. This interpretation is plausible given the task’s decision demands; however, readiness potentials typically emerge before the motor response.
Please clarify whether the timing of participants’ responses was considered when interpreting this negativity. In Figure 3 (and related figures), specify whether time 0 refers to stimulus presentation or movement onset, and discuss whether the potential occurs before or after the actual response. A clearer graphical indication of this temporal alignment would strengthen the argument.
Lines 482–485:
Later, the authors state that no early negative difference (i.e., readiness potential) was observed in response-locked trials. This appears inconsistent with the previous interpretation. Please justify in more detail why a post-stimulus negative ERP is interpreted as preparatory activity, and reconcile this explanation with the absence of a corresponding pre-movement component.
Figure 7:
In Figure 7, there seems to be a small ERP deflection immediately before stimulus presentation that could correspond to preparatory activity. Have the authors evaluated this component? If so, it could be valuable to mention briefly in the Discussion, as it might support a more classical readiness-potential interpretation.
Conclusion
Topographic findings:
The manuscript reports significant clusters in specific electrode regions but does not discuss why these particular topographies (e.g., parietal–occipital or posterior sites) might be functionally relevant. Given that the Conclusion emphasizes the importance of topographical analysis (lines 563–564), the Discussion could be strengthened by briefly interpreting the functional implications of the observed scalp distributions (e.g., posterior involvement in visual processing or motor preparation).
Author Response
Introduction
Lines 74–76: “Ultimately, approach–avoidance behaviors reflect a balance between explicit, emotion-driven, and implicit, reflective mechanisms…”
It seems that the terms explicit and implicit have been reversed here. According to earlier lines, explicit processes are reflective and deliberative, whereas implicit ones are emotion-driven and automatic. I suggest correcting this sentence for conceptual consistency.
DONE. We appreciate the reviewer’s detailed revision of the manuscript. We have checked and corrected the term inversion for clarity.
Methods
Lines 139–140: “Nonetheless, we acknowledge the ongoing debate about the symbolic meaning of these movements, as addressed in detail by Grasso-Cladera et al. [45].”
This is an important conceptual limitation. It could be discussed in more depth in the Methods or Limitations section, clarifying how the authors handled or mitigated the symbolic ambiguity of push–pull movements (e.g., by including the zoom effect, explicit instructions, or task familiarization).
DONE. The authors thank the reviewer for raising the need for further discussion of the symbolic meaning of the push-and-pull movements. We have added a paragraph in the Limitations section that addresses this issue and explains how we account for it in our experiment.
Lines 176–179: “Before the picture trials, participants performed 20 no-stimulus trials in which they were instructed to push or pull the joystick whenever the fixation cross changed color. This was done to obtain baseline data while using the joystick and to help participants familiarize themselves with the movement”
The paragraph appears twice. Please remove the duplicate section.
DONE. We thank the reviewer for pointing this out. We have removed the duplicated paragraph.
Discussion
Lines 475–479:
The authors propose that the observed negative potential may reflect a readiness potential. This interpretation is plausible given the task’s decision demands; however, readiness potentials typically emerge before the motor response.
Please clarify whether the timing of participants’ responses was considered when interpreting this negativity. In Figure 3 (and related figures), specify whether time 0 refers to stimulus presentation or movement onset, and discuss whether the potential occurs before or after the actual response. A clearer graphical indication of this temporal alignment would strengthen the argument.
DONE. We have adjusted the paragraph to clarify that the timing most likely falls before the response. However, we also rewrote the phrasing to be more tentative in our interpretation. We have additionally created Figure 1C to highlight the temporal alignments and included the description in the following figures, including Figure 3.
Lines 482–485:
Later, the authors state that no early negative difference (i.e., readiness potential) was observed in response-locked trials. This appears inconsistent with the previous interpretation. Please justify in more detail why a post-stimulus negative ERP is interpreted as preparatory activity, and reconcile this explanation with the absence of a corresponding pre-movement component.
DONE. We thank the reviewer for bringing this to our attention. We have reworked the paragraph to better reflect that. While one can speculate on the contributions of this activity, it is hard to draw a definite conclusion given the limited literature on the field and the contrasting findings reported in previous work.
Figure 7:
In Figure 7, there seems to be a small ERP deflection immediately before stimulus presentation that could correspond to preparatory activity. Have the authors evaluated this component? If so, it could be valuable to mention briefly in the Discussion, as it might support a more classical readiness-potential interpretation.
ADDRESSED. We improved the plot to clarify that the ERP is locked to movement onset rather than stimulus onset. We hope that the modification and the inclusion of panel C in Figure 1 make the interpretation easier.
Conclusion
Topographic findings:
The manuscript reports significant clusters in specific electrode regions but does not discuss why these particular topographies (e.g., parietal–occipital or posterior sites) might be functionally relevant. Given that the Conclusion emphasizes the importance of topographical analysis (lines 563–564), the Discussion could be strengthened by briefly interpreting the functional implications of the observed scalp distributions (e.g., posterior involvement in visual processing or motor preparation).
DONE. The authors thank the reviewer for the suggestion. We have incorporated information about the potential functional processes underlying the topographical map of activity for each one of the significant analyses.
Reviewer 2 Report
Comments and Suggestions for Authors
This study employs electroencephalography (EEG) to systematically investigate the neural dynamics of approach-avoidance bias (AAB) in a healthy population, focusing on stimulus-locked and response-locked event-related potential (ERP) components and frontal alpha asymmetry (FAA). The research design is rigorous, and the data analysis methods are advanced, particularly the use of threshold-free cluster enhancement (TFCE) for multiple comparison correction in ERP analysis, which represents a methodological strength. However, the study has certain limitations in terms of innovation, experimental validation of core hypotheses, and the depth of result interpretation.
-
The ERP analysis did not reveal a main effect of Condition (congruent vs. incongruent), which is partially inconsistent with the study's original intent. This suggests that EEG may not have effectively captured the core neural mechanisms of cognitive control within the AAB.
-
The study aimed to reveal the neural dynamics of AAB, particularly the neural basis of condition differences. However, the absence of a main effect of Condition in the ERP analysis means the core hypothesis was not sufficiently validated.
-
The interpretation of null results (e.g., no significant FAA differences) is somewhat generalized and lacks in-depth discussion of their methodological or theoretical implications.
-
There is a lack of comparison with traditional methods or baseline models, such as comparing analyses with and without TFCE, or contrasting results with other neuroimaging methods like fMRI.
-
Future research could integrate time-frequency analysis and deep learning models to better capture the non-linear, high-dimensional neural dynamics involved in AAB. For instance, leveraging wavelet time-frequency attention mechanisms (DOI: 10.1016/j.asoc.2025.113522) to enhance critical features, while drawing on the advantages demonstrated by methods like Generative Adversarial Networks with Non-negative Tensor Decomposition (F-GAN-NTD) in handling high-dimensional and incomplete neuroimaging data, could provide a new analytical framework for AAB research.
-
Ablation experiments were not conducted, such as investigating whether the number of trials was insufficient or whether preprocessing steps (e.g., ASR, ICA) influenced the results.
Author Response
- The ERP analysis did not reveal a main effect of Condition (congruent vs. incongruent),
which is partially inconsistent with the study's original intent. This suggests that EEG may
not have effectively captured the core neural mechanisms of cognitive control within the
AAB.
DONE. We thank the reviewer for pointing this out to our attention. We have added a paragraph in the Limitations section on the sensitivity of EEG for measuring activity at the subcortical level and its implications for our study and future research.
- The study aimed to reveal the neural dynamics of AAB, particularly the neural basis of
condition differences. However, the absence of a main effect of Condition in the ERP analysis means the core hypothesis was not sufficiently validated.
ADDRESSED. We thank the reviewer for pointing this out. We have explicitly indicated the initial expectation of the present study in the Introduction section. Furthermore, we have addressed this point in the Limitations section.
- The interpretation of null results (e.g., no significant FAA differences) is somewhat
generalized and lacks in-depth discussion of their methodological or theoretical implications.
DONE. We appreciate the suggestion to improve the interpretation of the results. We have modified the paragraph, adding information regarding the methodological limitations and implications of the null results.
- There is a lack of comparison with traditional methods or baseline models, such as comparing analyses with and without TFCE, or contrasting results with other neuroimaging methods like fMRI.
DONE. The authors thank the reviewer for mentioning the need for further explanation on the selected analytical strategy. We have modified the EEG analysis section to provide evidence of alignment between the study's objective and the analytical approach. We also included a discussion regarding other methods in the Limitations section.
- Future research could integrate time-frequency analysis and deep learning models to better capture the non-linear, high-dimensional neural dynamics involved in AAB. For instance, leveraging wavelet time-frequency attention mechanisms (DOI: 10.1016/j.asoc.2025.113522) to enhance critical features, while drawing on the advantages demonstrated by methods like Generative Adversarial Networks with Non-negative Tensor Decomposition (F-GAN-NTD) in handling high-dimensional and incomplete neuroimaging data, could provide a new analytical framework for AAB research.
DONE. We appreciate the reviewer’s suggestion. We have incorporated a paragraph in the Limitations section regarding future lines of research and the combination of methodologies to study the modulation of the AAB on brain activity.
- Ablation experiments were not conducted, such as investigating whether the number of trials was insufficient or whether preprocessing steps (e.g., ASR, ICA) influenced the results.
DONE. We thank the reviewer for raising concerns regarding potential influences and limitations of our results. We have provided evidence for the preprocessing steps conducted and added these points to the Limitations section.
Reviewer 3 Report
Comments and Suggestions for Authors
This manuscript presents a well-designed and clearly written EEG study investigating the neural correlates of the Approach–Avoidance Bias (AAB) in a non-clinical sample. The work is methodologically rigorous, based on relevant literature, and offers new insights into the temporal and spatial features of affect-driven decision-making. The dual focus on stimulus-locked and response-locked ERPs, complemented by frontal alpha asymmetry (FAA) analysis, represents a notable strength. The key advantage of this paper is that the study uses a modern and well-documented EEG preprocessing pipeline, ensuring reproducibility and signal integrity. Besides the transparency and data and code availability via GitHub enhances replicability.
The paper is of high quality and suitable for publication in Brain Sciences, following are minor revisions.
Comments:
1- Each condition contained only a low number of trials. This low trial number may limit the reliability of ERP results. Please include a brief discussion of the statistical power or signal-to-noise implications.
2-Condition factor (congruent vs. incongruent), the null finding for Condition deserves deeper discussion. Could the lack of effect stem from EEG poor sensitivity to subcortical sources (amygdala or basal ganglia)? The discussion mentions this but could be expanded with citations to studies showing this limitation.
3-Clarify the meaning of “brain dynamics, the title and abstract refer to “brain dynamics,” yet analyses focus primarily on ERP amplitude and timing. Consider clarifying that this term refers to temporal and spatial patterns of cortical activation, to avoid overgeneralization.
4-Lines 172-175 and lines 176 –179, is duplicated. Before the picture trials, participants performed 20 no-stimulus trials in which were instructed to push or pull the joystick whenever the fixation cross changed color. This was done to obtain baseline data while using the joystick and to help participants familiarize themselves with the movement.
5-Figures 3–8 could include TFCE threshold levels or a clearer explanation of color scales. This would improve interpretability for non-expert readers.
6-Include a schematic summary figure showing the timeline of the stimulus-locked and response-locked ERP windows and corresponding significant clusters (e.g., a graphical abstract would be great).
7-The author could enrich the introduction by citing these papers citations specially in the feature extraction part and time domain extracted features:
1- Orban, M., Zhang, X., Lu, Z., Zhang, Y., & Li, H. (2019, December). An Approach for Accurate Pattern Recognition of Four Hand Gestures Based on sEMG Signals. In Proceedings of the 2019 2nd International Conference on Control and Robot Technology (pp. 145-150).
Author Response
Comments:
1- Each condition contained only a low number of trials. This low trial number may limit the reliability of ERP results. Please include a brief discussion of the statistical power or signal-to-noise implications.
DONE. We included in the Limitations section a discussion of the number of trials per condition and its implications for signal-to-noise ratio and statistical power.
Line 659-670:
Similarly, some limitations of the EEG data should be acknowledged to ensure a cautious interpretation of the results and to highlight directions for future work. To start, our experiment had a relatively small number of trials per condition (N = 20). Because EEG signals are inherently noisy, averaging across trials is essential to increase the signal-to-noise ratio (SNR) and obtain reliable estimates of ERPs [112,113]. Previous research has shown that the reliability of ERP measures increases substantially with trial count, particularly for components with lower amplitudes or more variable latencies [114,115]. With a limited number of trials, as in our case, residual noise may therefore contribute to increased variability across participants and reduced statistical power to detect effects [114,116]. Future studies should include more trials per condition to enhance SNR and improve the robustness and reproducibility of the results.
2-Condition factor (congruent vs. incongruent), the null finding for Condition deserves deeper discussion. Could the lack of effect stem from EEG poor sensitivity to subcortical sources (amygdala or basal ganglia)? The discussion mentions this but could be expanded with citations to studies showing this limitation.
DONE. We thank the reviewer for pointing this out to our attention. We have added a paragraph in the Limitations section on the sensitivity of EEG for measuring subcortical activity and its implications for our study and future research.
Line 671-690:
Furthermore, EEG, as a measurement technique, has lower spatial resolution in comparison with other neuroimaging techniques for the study of the AAB modulation on brain activity. This is especially relevant for the present study, since the primary structures involved in approach-avoidance tendencies are related to the limbic system and, therefore, are subcortical. Hence, the EEG measurement offers poor sensitivity to subcortical activity in regions associated with emotional processing and decision-making, as it can measure only cortical sources of activity. The interpretation of the present results must take these limitations into account, particularly when considering findings such as the absence of a significant effect of the Condition (e.g., congruent vs. incongruent). Although the data suggest that the different conditions do not modulate cortical activity detectable at the scalp level, this does not imply that no neural differences exist. Rather, such differences may simply not be accessible with EEG recordings. Given this limitation, future studies would benefit from incorporating complementary neuroimaging techniques (e.g., fNIRS and fMRI) to examine the brain dynamics of the AAB. Furthermore, incorporating other interventions along with EEG measurement, such as transcranial focused ultrasound (tFUS) [117–119], may provide deeper insights into the effects of the AAB on neural dynamics. Together, acknowledging the limitations of EEG and supporting these shortcomings with other strategies would advance the study and interpretation of the AAB modulation at the subcortical level.
3-Clarify the meaning of “brain dynamics, the title and abstract refer to “brain dynamics,” yet analyses focus primarily on ERP amplitude and timing. Consider clarifying that this term refers to temporal and spatial patterns of cortical activation, to avoid overgeneralization.
DONE. The authors appreciate the request for clarification. We have included in the aim of the study a description of what we mean by “brain dynamics”.
Line 83-90:
The present exploratory study uses EEG to systematically examine brain dynamics in healthy participants performing a classic Approach–Avoidance Task (AAT), focusing on Event-Related Potentials (ERPs) and changes in frontal alpha synchronization (Frontal Alpha Asymmetry; FAA). To this end, we partially replicated the experimental paradigm implemented by Solzbacher and colleagues [16] and collected EEG data while participants performed the AAT. Following this approach, we aim to study cortical processes related to emotional decision making by using EEG data.
4-Lines 172-175 and lines 176 –179, is duplicated. Before the picture trials, participants performed 20 no-stimulus trials in which were instructed to push or pull the joystick whenever the fixation cross changed color. This was done to obtain baseline data while using the joystick and to help participants familiarize themselves with the movement.
DONE. We thank the reviewer for pointing this out. We have removed the duplicated paragraph.
5-Figures 3–8 could include TFCE threshold levels or a clearer explanation of color scales. This would improve interpretability for non-expert readers.
DONE. We appreciate the suggestion to improve interpretability. We have added a description in the figure caption for all topographical figures to clarify when we are discussing differences in voltage and when we refer to statistical values (F values). Line 385-390; Line 403-410; Line 428-437; Line 445-449; Line 468-447; Line 486-491;
6-Include a schematic summary figure showing the timeline of the stimulus-locked and response-locked ERP windows and corresponding significant clusters (e.g., a graphical abstract would be great).
DONE. The authors thank the reviewer for the suggestion to clarify the experimental design and the posterior analyses. We have modified Figure 1 (the experimental task) to include information on the temporal dimension of task-relevant events. Line 182-191
7-The author could enrich the introduction by citing these papers citations specially in the feature extraction part and time domain extracted features:
Orban, M., Zhang, X., Lu, Z., Zhang, Y., & Li, H. (2019, December). An Approach for Accurate Pattern Recognition of Four Hand Gestures Based on sEMG Signals. In Proceedings of the 2019 2nd International Conference on Control and Robot Technology (pp. 145-150).
DONE. We appreciated the reviewers suggestion to improve the manuscript, we have incorporated the citation in the Limitations section. References 115
Round 2
Reviewer 1 Report
Comments and Suggestions for Authors
The authors have responded to all my comments and adapted the manuscript accordingly, and I have no further suggestions to add.
Reviewer 2 Report
Comments and Suggestions for Authors
-
The authors have not thoroughly addressed the previous review comments by adding necessary experiments or theoretical support. This has resulted in significant shortcomings in the innovation and experimental comparisons of the current revised version. It is recommended to submit a properly revised version after careful modifications and not to waste my time. In your response, you must list the specific changes made in the point-by-point response letter, rather than simply asking me to locate them in the manuscript.
-
Differences were found for the Valence factor in early P100 and late LPP-like components, supporting the view of early emotional processing and motor preparation. However, no significant ERP differences were found for the Condition factor (congruent vs. incongruent), which is inconsistent with expectations and weakens the validation of the neural mechanisms underlying "cognitive control."
-
No control conditions, such as those "without the zoom effect" or "without movement requirements," were implemented, making it difficult to dissociate the contributions of motor execution from cognitive decision-making.
-
The exploration of different frequency bands (e.g., theta, beta) was not conducted, with the analysis focusing solely on alpha asymmetry, which appears somewhat limited.
-
The peak amplitude and latency statistics for ERP components were not reported. Relying exclusively on cluster-based analysis limits the interpretative depth of the results.
Author Response
- The authors have not thoroughly addressed the previous review comments by adding necessary experiments or theoretical support. This has resulted in significant shortcomings in the innovation and experimental comparisons of the current revised version. It is recommended to submit a properly revised version after careful modifications and not to waste my time. In your response, you must list the specific changes made in the point-by-point response letter, rather than simply asking me to locate them in the manuscript.
We apologize for the missing line numbers; something went wrong during the upload of the point-to-point reply. We have added to this review round the lines where changes were made in the previous round, along with the textual citation for each change.
- Differences were found for the Valence factor in early P100 and late LPP-like components, supporting the view of early emotional processing and motor preparation. However, no significant ERP differences were found for the Condition factor (congruent vs. incongruent), which is inconsistent with expectations and weakens the validation of the neural mechanisms underlying "cognitive control."
We appreciate the reviewer’s comment on this remark. The present study employed an exploratory design to investigate changes in brain activity during a classic Approach-Avoidance Task. We have addressed in the discussion section (Lines 547 to 555) what we saw in the data and the limitations (Lines 684 to 703) in interpreting the results. We agreed with the reviewer, as we had some initial expectations about finding differences for the condition factor; however, this was overruled by the data.
- No control conditions, such as those "without the zoom effect" or "without movement requirements," were implemented, making it difficult to dissociate the contributions of motor execution from cognitive decision-making.
We thank the reviewer for pointing out the absence of a control condition without the zoom effect. We implemented the zooming effect because it has been used in several studies and is reliable for eliciting the AAB. Furthermore, the AAB includes a motor component (approach or avoid); hence, we opted to follow a traditional approach and use a task that requires a motor response. As pointed out by the reviewer, the absence of a control condition without the zoom effect can affect the interpretation of the results, and we have added a remark in the discussion section about future directions (Lines 565 to 576), which now reads as follows:
“The absence of such an early negativity, combined with the stimulus-locked findings, complicates a straightforward interpretation of readiness potential differences. Instead, we believe the difference we observe can be explained by the task's properties, such as the stimuli's zooming effect, which aligns with our results' temporal and spatial characteristics. Future work can rule out this effect by comparing the data from two experimental tasks, with and without the addition of the zoom effect. To our knowledge, this study is one of the first attempts to systematically explore movement onset ERPs in the context of the AAB. Therefore, drawing definitive conclusions about the found clusters and their attributions to different processes, such as the readiness potential, is currently only speculative. Further research is needed to disentangle the contributions of valence to movement-related ERP differences.”
Also, we have incorporated a paragraph on discussion methodological considerations related to the AAT in the Limitations section (Lines 650 to 671), which reads:
“While our study provides insights into the brain dynamics of the AAB, certain methodological and analytical constraints need to be considered. First, regarding the experimental design of the task, it is worth noting that our results are consistent with the standard view of the AAB as a distance‐regulation mechanism, in which changes in proximity between an agent and an object underlie approach–avoidance behavior [11,16,105]. However, although pushing and pulling movements are commonly used to operationalize the bias in both desktop and VR tasks [9,16,106], their interpretation is not straightforward [107]. These actions do not inherently map onto approach or avoidance, as the same movement can serve opposite functions depending on context and stimulus properties [108–111]. To mitigate the effect of the symbolic meaning of pushing and pulling, we implemented a series of methodological considerations. First, the “zoom effect” (i.e., increasing the size of the picture when pulling and decreasing its size when pushing) after the movement performed by the subject served as a way to augment the impression of an approach or avoidance behavior. Furthermore, we repeated the presentation of the instructions after a regular number of trials, in order to prevent confusion regarding the type of block (i.e., congruent or incongruent) and the type of movement required for each valence in the current trial. Lastly, we incorporated a series of test trials at the beginning of each block as a way to generate familiarity with the task and the required movements for approaching and avoiding. Overall, regarding the open debate about the symbolic meaning of the movements used to study approach and avoidance behaviors, our experimental design attempted to mitigate its effect on the results.”
- The exploration of different frequency bands (e.g., theta, beta) was not conducted, with the analysis focusing solely on alpha asymmetry, which appears somewhat limited.
We acknowledge the reviewer’s concern regarding our methodological decision to focus on the FAA. Our analytical approach followed a conservative rationale, consistent with previous EEG studies on the AAB, which have primarily examined FAA in the context of frequency-domain analyses.. To our knowledge, there is currently no empirical evidence supporting the relevance or inclusion of other frequency bands in this specific framework.
- The peak amplitude and latency statistics for ERP components were not reported. Relying exclusively on cluster-based analysis limits the interpretative depth of the results.
The authors thank the reviewer for the suggestions to improve the report of our data. We have added a report of peak amplitude and latency for the grand average of the main electrodes (e.g., the peak of each cluster) (Lines 375 to 389), and we have improved the caption of Figure 3 (Lines 394 to 401). The mentioned result section now reads:
“For stimulus-locked analyses, we found the peak for the significant cluster for the Valence factor at 86 ms after stimulus onset at channel M2, while for the interaction of factors, the peak was at 582 ms after stimulus onset at channel P6. The average ERP over conditions at M2 showed a highest peak of 3.056 μV at ~130 ms post picture onset, and a lowest peak (-0.778 μV) at ~180 ms post stimulus onset. At channel P6, the average over conditions showed a positive peak of 7.891 μV at ~125 ms after picture presentation, and a negative deflection (-4.398 μV) with a peak at ~20 ms post stimulus onset. For response-locked ERPs, we found the peak of the cluster for the Valence factor at 2 ms after movement onset at channel P5 and 108 ms post movement onset at channel POz for the interaction of factors. The average ERP over conditions at channel P5 presented a positive peak at ~ -30 ms previous to movement onset (1.901 μV) and a negative peak of -2.034 at ~130 ms after movement onset. Similarly, at channel POz, the average ERP across conditions showed a positive deflection of 2.861 μV at ~ -30 ms before movement onset, and a negative peak at ~130 ms after movement onset (-4.901 μV).”
Figure caption:
“Figure 3. ERPs for all peaks of significant clusters. Panel A represents stimulus-onset ERPs (locked to picture onset; red line, Figure 1C) of all target conditions for the Valence factor, given its statistical significance. Panel B shows the stimulus-onset ERPs of all target conditions for the interaction of the Valence and Condition factors, given its statistical significance. Similarly, panels C and D display movement onset ERPs (locked to movement onset; purple line, Figure 1C) for Valence and interaction of factors, respectively. The gray line represents the temporal peak of the cluster. For visualization purposes, we have shortened the temporal window to -400 to 600 ms for picture onset trials and -500 to 400 ms for movement onset trials.”
Round 1, reviews
- The ERP analysis did not reveal a main effect of Condition (congruent vs. incongruent), which is partially inconsistent with the study's original intent. This suggests that EEG may not have effectively captured the core neural mechanisms of cognitive control within the AAB.
DONE [Lines 604 to 703]. We thank the reviewer for pointing this out to our attention. We have added a paragraph in the Limitations section on the sensitivity of EEG for measuring activity at the subcortical level and its implications for our study and future research. The section reads:
“Furthermore, EEG, as a measurement technique, has lower spatial resolution in comparison with other neuroimaging techniques for the study of the AAB modulation on brain activity. This is especially relevant for the present study, since the primary structures involved in approach-avoidance tendencies are related to the limbic system and, therefore, are subcortical. Hence, the EEG measurement offers poor sensitivity to subcortical activity in regions associated with emotional processing and decision-making, as it can measure only cortical sources of activity. The interpretation of the present results must take these limitations into account, particularly when considering findings such as the absence of a significant effect of the Condition (e.g., congruent vs. incongruent). Although the data suggest that the different conditions do not modulate cortical activity detectable at the scalp level, this does not imply that no neural differences exist. Rather, such differences may simply not be accessible with EEG recordings. Given this limitation, future studies would benefit from incorporating complementary neuroimaging techniques (e.g., fNIRS and fMRI) to examine the brain dynamics of the AAB. Furthermore, incorporating other interventions along with EEG measurement, such as transcranial focused ultrasound (tFUS) [117–119], may provide deeper insights into the effects of the AAB on neural dynamics. Together, acknowledging the limitations of EEG and supporting these shortcomings with other strategies would advance the study and interpretation of the AAB modulation at the subcortical level.”
- The study aimed to reveal the neural dynamics of AAB, particularly the neural basis of condition differences. However, the absence of a main effect of Condition in the ERP analysis means the core hypothesis was not sufficiently validated.
ADDRESSED [Lines 604 to 703]. We thank the reviewer for pointing this out. We have explicitly indicated the initial scope of the present study in the Introduction section. Furthermore, we have addressed this point in the Limitations section. The section reads:
“Furthermore, EEG, as a measurement technique, has lower spatial resolution in comparison with other neuroimaging techniques for the study of the AAB modulation on brain activity. This is especially relevant for the present study, since the primary structures involved in approach-avoidance tendencies are related to the limbic system and, therefore, are subcortical. Hence, the EEG measurement offers poor sensitivity to subcortical activity in regions associated with emotional processing and decision-making, as it can measure only cortical sources of activity. The interpretation of the present results must take these limitations into account, particularly when considering findings such as the absence of a significant effect of the Condition (e.g., congruent vs. incongruent). Although the data suggest that the different conditions do not modulate cortical activity detectable at the scalp level, this does not imply that no neural differences exist. Rather, such differences may simply not be accessible with EEG recordings. Given this limitation, future studies would benefit from incorporating complementary neuroimaging techniques (e.g., fNIRS and fMRI) to examine the brain dynamics of the AAB. Furthermore, incorporating other interventions along with EEG measurement, such as transcranial focused ultrasound (tFUS) [117–119], may provide deeper insights into the effects of the AAB on neural dynamics. Together, acknowledging the limitations of EEG and supporting these shortcomings with other strategies would advance the study and interpretation of the AAB modulation at the subcortical level.”
- The interpretation of null results (e.g., no significant FAA differences) is somewhat generalized and lacks in-depth discussion of their methodological or theoretical implications.
DONE [Lines 626 to 648]. We appreciate the suggestion to improve the interpretation of the results. We have modified the paragraph, adding information regarding the methodological limitations and implications of the null results. Now it reads:
“In contrast to the time-series analysis, our results showed no significant differences between FAA values for pleasant and unpleasant trials for stimulus-locked trials. Higher left hemispheric activity tends to be associated with positive and pleasurable stimuli, while higher right activity has been associated with negative valenced stimuli [86,87]. Given the characteristics of FAA, it has been a preferred measure to explore activity related to affective stimuli and motivation [94,95]; nevertheless, results are heterogeneous. The review conducted by Sabu and colleagues [95] shows that out of 18 studies using emotionally valenced pictures to assess differences in FAA, only two found an effect of the presented stimuli [96,97]. However, studies implementing videos [98,99], real cues [100,101], and games [102,103] were more prone to finding FAA differences. These results highlight a relevant methodological point regarding the suitability of static images for generating emotional engagement in the subjects and eliciting differences in FAA activity, especially when considering the positive results found by implementing relatable real-world cues or more immersive experiences by using videos, games, and even 3D stimuli and virtual environments [95,104,105]. In this sense, the consistent finding of limited FAA effects with static images across multiple studies suggests a systematic influence of stimulus type. This consistent lack of effect when using static stimuli likely explains the absence of a difference in the current study, suggesting that FAA may not be the most suitable measure in classical AAB experiments using picture stimuli. Instead, the heterogeneity in FAA results regarding emotionally valenced stimuli, and the high prevalence of negative results when using static stimuli or 2D pictures, posit the need for new methodological approaches for studying the AAB under more emotionally engaging and naturalistic scenarios.”
- There is a lack of comparison with traditional methods or baseline models, such as comparing analyses with and without TFCE, or contrasting results with other neuroimaging methods like fMRI.
DONE [Lines 279 to 285]. The authors thank the reviewer for mentioning the need for further explanation on the selected analytical strategy. We have modified the EEG analysis section to provide evidence of alignment between the study's objective and the analytical approach. This reads:
“To explore differences across conditions at all electrodes and time points, we performed a two-factor repeated measures ANOVA (2x2: Condition x Valence), with a significance level set at .05. To address the multiple comparison problems, we implemented a cluster-based permutation test using threshold-free cluster enhancement (TFCE) as described on the ept_TFCE toolbox [71]. We adopted this methodological strategy given the exploratory nature of our study, prioritizing a robust and unbiased data-driven approach [71].”
We also included a discussion regarding other methods in the Limitations section [Lines 696 to 703], which now reads:
“Given this limitation, future studies would benefit from incorporating complementary neuroimaging techniques (e.g., fNIRS and fMRI) to examine the brain dynamics of the AAB. Furthermore, incorporating other interventions along with EEG measurement, such as transcranial focused ultrasound (tFUS) [117–119], may provide deeper insights into the effects of the AAB on neural dynamics. Together, acknowledging the limitations of EEG and supporting these shortcomings with other strategies would advance the study and interpretation of the AAB modulation at the subcortical level.”
- Future research could integrate time-frequency analysis and deep learning models to better capture the non-linear, high-dimensional neural dynamics involved in AAB. For instance, leveraging wavelet time-frequency attention mechanisms (DOI: 10.1016/j.asoc.2025.113522) to enhance critical features, while drawing on the advantages demonstrated by methods like Generative Adversarial Networks with Non-negative Tensor Decomposition (F-GAN-NTD) in handling high-dimensional and incomplete neuroimaging data, could provide a new analytical framework for AAB research.
DONE [Lines 696 to 703]. We appreciate the reviewer’s suggestion. We have incorporated a paragraph in the Limitations section regarding future lines of research and the combination of methodologies to study the modulation of the AAB on brain activity. The section reads:
“Given this limitation, future studies would benefit from incorporating complementary neuroimaging techniques (e.g., fNIRS and fMRI) to examine the brain dynamics of the AAB. Furthermore, incorporating other interventions along with EEG measurement, such as transcranial focused ultrasound (tFUS) [117–119], may provide deeper insights into the effects of the AAB on neural dynamics. Together, acknowledging the limitations of EEG and supporting these shortcomings with other strategies would advance the study and interpretation of the AAB modulation at the subcortical level.”
- Ablation experiments were not conducted, such as investigating whether the number of trials was insufficient or whether preprocessing steps (e.g., ASR, ICA) influenced the results.
DONE [Lines 218 to 225]. We thank the reviewer for raising concerns regarding potential influences and limitations of our results. We have provided evidence for the preprocessing steps conducted. The section now reads:
“EEG data preprocessing was conducted in the MATLAB (R2024b) environment using EEGLAB [54] (version 2024.1) with a custom script adjusted from Schmidt and Nolte [55]. The implemented preprocessing routine was developed following advanced preprocessing guidelines for EEG data and previous work in the field [56–58]. First, the data was imported into MATLAB, ensuring double precision for all preprocessing steps [59]. Non-empirical segments (e.g., pre-task intervals) were removed. Channel labels were standardized to the 10-5 BESA system, and channels with no recorded data were excluded.”
Furthermore, we have added these points to the Limitations section [Lines 672 to 683]. This reads:
“Similarly, some limitations of the EEG data should be acknowledged to ensure a cautious interpretation of the results and to highlight directions for future work. To start, our experiment had a relatively small number of trials per condition (N = 20). Because EEG signals are inherently noisy, averaging across trials is essential to increase the signal-to-noise ratio (SNR) and obtain reliable estimates of ERPs [112,113]. Previous research has shown that the reliability of ERP measures increases substantially with trial count, particularly for components with lower amplitudes or more variable latencies [114,115]. With a limited number of trials, as in our case, residual noise may therefore contribute to increased variability across participants and reduced statistical power to detect effects [114,116]. Future studies should include more trials per condition to enhance SNR and improve the robustness and reproducibility of the results.”
Reviewer 3 Report
Comments and Suggestions for Authors
It’s ok. It’s not perfect but it can work. You can go forward with this version.
Author Response
We appreciate the reviewer's feedback.